# SHIP1 deficiency causes inflammation-dependent retardation in skeletal growth

Fatemeh Safari[1,2,5], Wen Jie Yeoh[2,3], Saskia Perret-Gentil[1], Frank Klenke[4], Silvia Dolder[1], Willy Hofstetter[1,6,*], Philippe Krebs[3,*]

Inflammation and skeletal homeostasis are closely intertwined. Inflammatory diseases are associated with local and systemic bone loss, and post-menopausal osteoporosis is linked to low-level chronic inflammation. Phosphoinositide-3-kinase signalling is a pivotal pathway modulating immune responses and controlling skeletal health. Mice deficient in Src homology 2–containing inositol phosphatase 1 (SHIP1), a negative regulator of the phosphoinositide-3-kinase pathway, develop systemic inflammation associated with low body weight, reduced bone mass, and changes in bone microarchitecture. To elucidate the specific role of the immune system in skeletal development, a genetic approach was used to characterise the contribution of SHIP1-controlled systemic inflammation to SHIP1-dependent osteoclastogenesis. Lymphocyte deletion entirely rescued the skeletal phenotype in $Rag2^{-/-}/Il2rg^{-/-}/SHIP1^{-/-}$ mice. $Rag2^{-/-}/Il2rg^{-/-}/SHIP1^{-/-}$ osteoclasts, however, displayed an intermediate transcriptomic signature between control and $Rag2^{+/+}/Il2rg^{+/+}/SHIP1^{-/-}$ osteoclasts while exhibiting aberrant in vitro development and functions similar to $Rag2^{+/+}/Il2rg^{+/+}/SHIP1^{-/-}$ osteoclasts. These data establish a cell-intrinsic role for SHIP1 in osteoclasts, with inflammation as the key driver of the skeletal phenotype in SHIP1-deficient mice. Our findings demonstrate the central role of the immune system in steering physiological skeletal development.

## Introduction

Bone, as the site of haematopoiesis after birth, has a close anatomical and functional relation with marrow cells and their secretome. Immune cells, which are a part of the bone marrow and dictate inflammatory processes, exert profound effects on skeletal metabolism. For example, after trauma or surgery, an inflammatory phase in which blood clots and debris are removed from the repair site is necessary to initiate bone repair. Conversely, increased immune cell-mediated secretion of pro-inflammatory cytokines such as interleukin (IL)-1$\beta$, IL-6, and TNF contribute to bone loss and impaired bone formation in chronic inflammatory disorders [1]. Indeed, numerous diseases with underlying chronic inflammatory processes, such as rheumatoid arthritis (RA), inflammatory bowel disease, periodontitis, and chronic obstructive pulmonary disease, are associated with changes in skeletal metabolism [1, 2]. Furthermore, the development of age-related degenerative diseases, including post-menopausal osteoporosis, is promoted by chronic low-grade inflammatory processes [3, 4]. Despite the availability of a panel of anti-catabolic and anabolic drugs for treating osteoporosis, each of these therapeutic measures has limitations, and a better understanding of the contribution of inflammatory processes to bone loss is required to develop new targeted therapeutic approaches [5, 6, 7].

Bone is a dynamic organ that is continuously remodelled. To maintain bone mass and architecture, bone formation by osteoblasts (OB) and bone resorption by osteoclasts (OC) must be tightly coordinated during modelling and remodelling processes. In addition, both innate and adaptive immune cells modulate the development and activity of bone cell lineages in health and disease. In particular, dendritic cells (DC), T/B cells, natural killer (NK) cells, and innate lymphoid cells affect bone turnover and architecture by releasing inflammatory mediators that modulate the formation and resorption processes [8].

B- and T-cells contribute to inflammatory bone loss in RA by releasing receptor activator of nuclear factor-$\kappa$B ligand (RANKL) and IL-17, which stimulate the development of osteoclasts [9, 10]. RANKL was first described as a T-cell factor that stimulates DC and modulates both B- and T-cell development [11]. Furthermore, as an essential factor for OC development and function, RANKL is a key regulator of bone remodelling [12]. In addition to exhibiting osteopetrosis due to the lack of OC, RANKL-deficient mice also display impaired lymphocyte development [13, 14]. Moreover, recent studies

---

[1]Bone & Joint Program, Department for BioMedical Research (DBMR), University of Bern, Bern, Switzerland  [2]Graduate School for Cellular and Biomedical Sciences, University of Bern, Bern, Switzerland  [3]Institute of Tissue Medicine and Pathology, University of Bern, Bern, Switzerland  [4]Department of Orthopaedic Surgery, Inselspital, Bern University Hospital, Bern, Switzerland  [5]AO Research Institute Davos, Davos, Switzerland  [6]Department of Cranio-Maxillofacial Surgery, Inselspital, Bern University Hospital, Bern, Switzerland

Correspondence: willy.hofstetter@unibe.ch; philippe.krebs@unibe.ch
*Willy Hofstetter and Philippe Krebs contributed equally to this work

have suggested that innate lymphoid cells play a role in modulating joint inflammation (15, 16, 17).

Besides RANKL, CSF-1 is the second essential growth factor in osteoclastogenesis. CSF-1 and RANKL, through their respective signalling pathways, activate phosphoinositide-3-kinase (PI3K)/AKT, which controls virtually all steps in OC development (18). The PI3K pathway is also critically involved in lymphoid and non-lymphoid cell development and activation (19), and dysregulation of this pathway has been observed in inflammatory diseases, autoimmunity, and cancer (19, 20). Several studies have reported the pivotal role of the PI3K pathway in the regulation of skeletal health (21, 22, 23, 24).

PI3K/AKT signalling is negatively regulated by several phosphatases, including phosphatase and tensin homologue (PTEN), Src homology 2-containing inositol phosphatase 1 (SHIP1) and SHIP2 (25). SHIP1, encoded by *INPP5D*, regulates PI3K/AKT signalling by hydrolysing the 5′-phosphate group of the product of PI3K, PI(3,4,5)P3, and converting it into PI(3,4)P2. Whereas PTEN and SHIP2 are ubiquitously expressed, SHIP1 expression is restricted mainly to haematopoietic cells. SHIP1-deficient mice develop a myeloproliferative disorder, resulting in progressive and fatal wasting disease caused by myeloid cell infiltration into several organs (26, 27). In addition, SHIP1 deficiency leads to a low body weight and bone mass phenotype, the latter being previously reported to be caused by an increase in the number of hyper-resorptive OC (28). However, selective deletion of SHIP1 in myeloid cells, including OC lineage cells, failed to affect bone mass and body weight (29). Other studies have shown that SHIP1 is also expressed in non-haematopoietic cells within the bone environment (30), and diminished bone mass and density in SHIP1-deficient mice have been proposed to result from SHIP1 deficiency in mesenchymal stem cells and OB progenitors (29). Therefore, the precise role of SHIP1 in inflammation-associated bone loss remains unclear.

In this study, we used a genetic approach to ablate lymphoid cells in SHIP1-deficient mice, thereby discriminating the contribution of SHIP1-regulated systemic inflammation from that of SHIP1-dependent osteoclastogenesis. Our findings suggest that SHIP1 expression in OC and OB plays a limited role in skeletal development under physiological conditions. However, in the context of inflammation, the interaction of lymphoid cells with bone cell lineages promotes skeletal alterations in SHIP1-deficient mice, highlighting the pivotal contribution of the immune system in the regulation of skeletal development.

# Results

## Genetic ablation of lymphoid cells corrects growth defects in SHIP1-deficient mice

SHIP1-deficient mice were used as models for chronic inflammatory diseases and bone pathology to study the contribution of systemic inflammation and the PI3K signalling pathway to skeletal development and maintenance. Specifically, we used the previously described *Inpp5d styx* mutant strain harbouring a point mutation resulting in the loss of SHIP1 protein expression (27). In comparison

with WT and $SHIP1^{+/styx}$ animals, SHIP1-deficient $SHIP1^{styx/styx}$ mice showed low body weight and reduced skeletal size at the age of 3 wk and later (Figs 1A and S1A and B). Genetic ablation of all lymphocyte populations in $SHIP1^{styx/styx}$ mice (by crossing them with animals lacking *Rag2* and *Il2rg*) restored body weight and bone size to WT levels in SHIP1-deficient mice (Figs 1B and S1A and B) and diminished the systemic levels of some, but not all, common inflammatory markers (Fig S1C–G). Analysis of the systemic levels of inflammatory mediators in the serum indicated that cytokine levels were either unchanged in the experimental groups (IL-1β and IL-6; Fig S1C and D), increased in SHIP1-deficient strains (TNF; Fig S1E), or increased in $SHIP1^{styx/styx}$ mice but were present at normal levels in $SHIP1^{styx/styx}$ animals lacking lymphocytes (MIP-1α and MIP-1β, the latter as a trend; Fig S1F and G).

Micro-computed tomography (micro-CT) analysis was performed to evaluate the microstructure of the distal femora and vertebrae (L4) in SHIP1-deficient mice with *Rag2/Il2rg*-proficient and *Rag2/Il2rg*-deficient backgrounds. Vertebral and femoral bone volume fractions (BV/TV) were lower in $SHIP1^{styx/styx}$ mice than in WT mice (40% and 30%, respectively) (Fig 1G and C). In the vertebrae, the trabecular number of $SHIP1^{styx/styx}$ mice increased by 27% (Fig 1H), whereas the thickness and spacing decreased by 33% and 20%, respectively (Fig 1I and J). In the femora from $SHIP1^{styx/styx}$ mice, trabecular thickness decreased by 20% (Fig 1E), whereas trabecular number and spacing remained unchanged (Fig 1D and F). Bone mass and microarchitecture were restored in $Rag2^{-/-}/Il2rg^{-/-}/SHIP1^{styx/styx}$ animals (Fig 1C–J).

Because SHIP1-deficient mice were characterised by a growth-retardation phenotype, micro-CT analysis of WT mice of different ages was performed to distinguish between growth delay and bone defects. The results showed changes in bone volume and trabecular microarchitecture during the observation period of up to 12 wk after birth. For both femora and vertebrae, BV/TV and trabecular thickness were correlated with age and size (Fig S1H, J, L, and N). However, the trabecular number and spacing remained unchanged (Fig S1I, K, M, and O).

Taken together, these data demonstrate that skeletal growth and architecture are impaired in $SHIP1^{styx/styx}$ mice and that this phenotype is reversed in $Rag2^{-/-}/Il2rg^{-/-}$ animals.

## OC-associated transcriptome signature is altered in bones from SHIP1-deficient mice

To assess the gene expression profile in bone tissues from SHIP1-deficient mice, transcriptome analysis of the femora of $SHIP1^{styx/styx}$ and $SHIP1^{+/+}$ mice with $Rag2^{-/-}/Il2rg^{-/-}$ and $Rag2^{+/+}/Il2rg^{+/+}$ backgrounds was performed. The analysis revealed that the expression of almost 2,000 genes was significantly altered in bones from SHIP1-deficient animals in comparison with WT bones, whereas only 55 genes were differentially expressed in $Rag2^{-/-}/Il2rg^{-/-}/SHIP1^{styx/styx}$ versus $Rag2^{-/-}/Il2rg^{-/-}/SHIP1^{+/+}$ (false discovery rate [FDR]-corrected *P*-value < 0.05, log$_2$-fold change > 1) femora (Fig 1K and L). The levels of transcripts encoding OC traits, such as calcitonin receptor (*Calcr*), matrix metalloproteinase 9 (*Mmp9*), and nuclear factor of activated T-cells (*Nfatc1*), were up-regulated exclusively in SHIP1-deficient $Rag2^{+/+}/Il2rg^{+/+}$ bones (Fig S2A–C and Tables S1 and S2). KEGG pathway analysis revealed that the WNT and TGF-β signalling

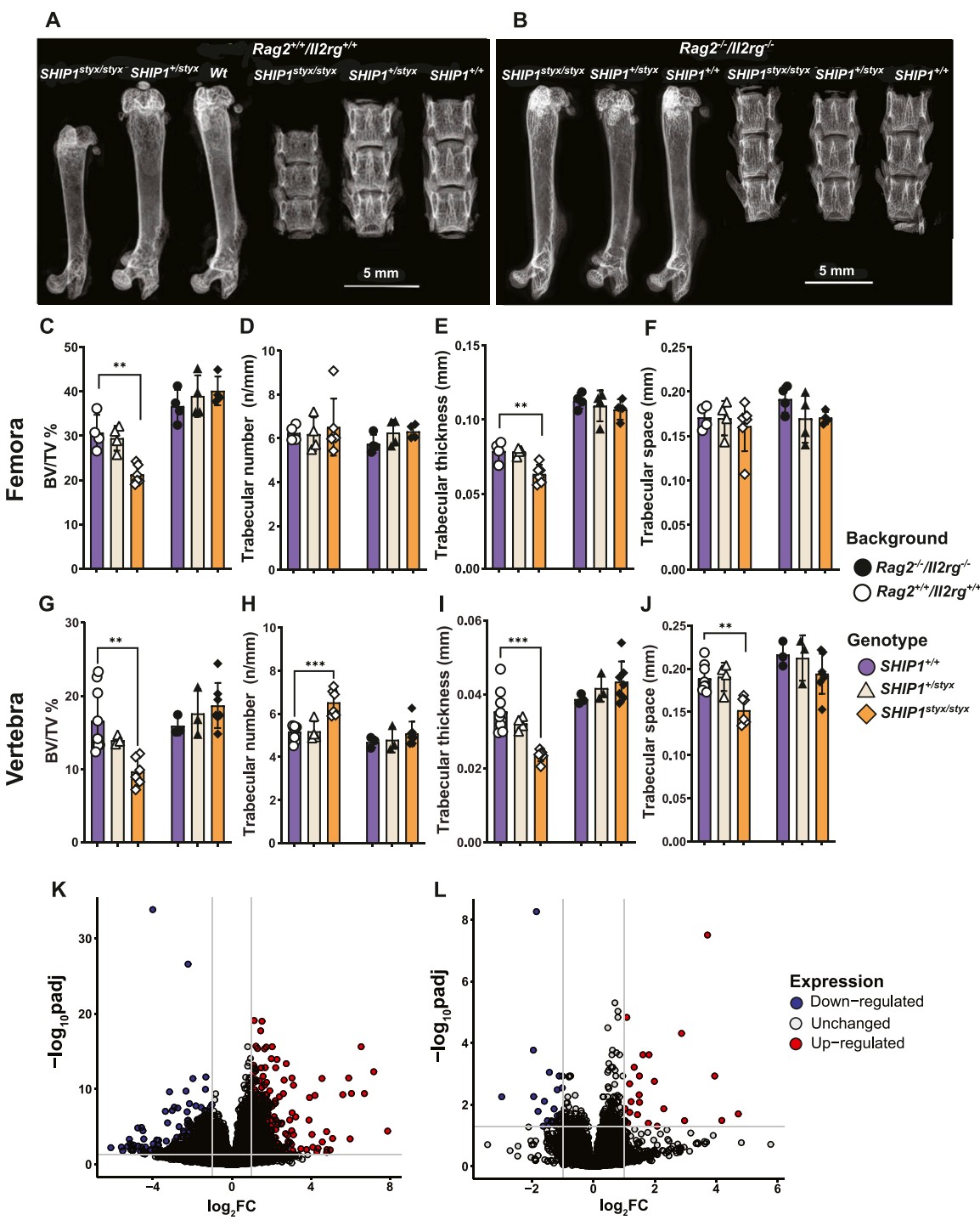

**Figure 1. Characterisation of the bones of SHIP1-deficient mice.**
**(A, B)** High-resolution X-Ray images of femora and vertebrae (L3–L5) of 9-wk-old mice *with Rag2^{+/+}/Il2rg^{+/+}* (A) background and *Rag2^{−/−}/Il2rg^{−/−}* (B) background.
**(C, D, E, F, G, H, I, J)** Micro-CT analysis of femoral (C, D, E, F) and vertebral (G, H, I, J) bone mass and architecture. *SHIP1^{styx/styx}* mice were characterised by a decrease in bone mass (BV/TV) in comparison with age-matched controls. Bone mass was restored in *SHIP1^{styx/styx}* mice bred on the *Rag2^{−/−}/Il2rg^{−/−}* background.
**(K, L)** Differentially expressed transcripts in SHIP1-deficient femora in comparison with controls. Transcriptome analysis was performed on diaphyseal bone of femora from each mouse strain. **(K, L)** Volcano plot representing the number of differentially expressed genes in *SHIP1^{styx/styx}* versus WT femora (K) and *Rag2^{−/−}/Il2rg^{−/−}/SHIP1^{styx/styx}* versus *Rag2^{−/−}/Il2rg^{−/−}/SHIP1^{+/+}* femora (L). Expression of many genes was altered in *SHIP1^{styx/styx}* femora in comparison with WT (FDR < 0.05 and log twofold change > 1). In *Rag2^{−/−}/Il2rg^{−/−}/SHIP1^{styx/styx}* femora, only few genes were differentially expressed in comparison with the controls. Data indicate mean ± SD of 3–9 mice per group. Significant changes were calculated with two-way ANOVA, **(P < 0.01), ***(P < 0.001), and ****(P < 0.0001).

pathways were down-regulated in SHIP1-deficient $Rag2^{+/+}/Il2rg^{+/+}$ femora, whereas the pathways regulating OC development were up-regulated (Fig S2D).

To assess the efficiency of gene ablation, the levels of mRNAs encoding SHIP1, RAG2, and IL2RG were determined using qRT-PCR. Levels of transcripts encoding SHIP1 were reduced by 90% and 50%, respectively, in $SHIP1^{styx/styx}$ and $SHIP1^{+/styx}$ OC, in both $Rag2^{-/-}/Il2rg^{-/-}$ and $Rag2^{+/+}/Il2rg^{+/+}$ backgrounds (Fig S3A). Transcripts encoding IL2RG were not detected in OC from $Rag2^{-/-}/Il2rg^{-/-}$ mice, in contrast to OC from $Rag2^{+/+}/Il2rg^{+/+}$ animals (Fig S3B). No transcripts encoding RAG2 were detectable in any of the OC genotypes (Fig S3C). Overall, the data showed that genetic ablation of lymphocytes restored the transcriptome landscape in the bones of SHIP1-deficient mice.

### SHIP1 modulates OC development in vitro

To address the possible intrinsic effects of SHIP1 deficiency on OC development, osteoclast progenitor cells (OPC) from $SHIP1^{styx/styx}$, $SHIP1^{+/styx}$, WT, $Rag2^{-/-}/Il2rg^{-/-}/SHIP1^{styx/styx}$, $Rag2^{-/-}/Il2rg^{-/-}/SHIP1^{+/styx}$, and $Rag2^{-/-}/Il2rg^{-/-}/SHIP1^{+/+}$ animals were cultured with CSF-1 and different doses of RANKL for 5 d. Cell proliferation assessed by XTT cell viability assay demonstrated an increase of 50% in cell number in cultures of $SHIP1^{styx/styx}$ OPC in comparison with the cells from the other genotypes, whereas the activity of tartrate-resistant acid phosphatase (TRAP), a differentiation marker of OC lineage cells, was similar in all cultures (Fig 2A and B). This resulted in a 35% decrease in TRAP/XTT in $SHIP1^{styx/styx}$ cultures (Figs 2C and S4). The number of cells and TRAP activity in cultures of $Rag2^{-/-}/Il2rg^{-/-}/SHIP1^{styx/styx}$ OPC were 18% higher than those in the $Rag2^{-/-}/Il2rg^{-/-}/SHIP1^{+/+}$ controls (Fig 2D and E); however, TRAP/XTT expression was the same for cultures from the three genotypes (Fig 2F). To assess OC morphology, the cultures were stained with TRAP. In cultures of $SHIP1^{styx/styx}$ and $Rag2^{-/-}/Il2rg^{-/-}/SHIP1^{styx/styx}$ OPC, respectively, the morphology of OC-like cells was characterised by increases in size and number of nuclei per OC in comparison to $SHIP1^{+/styx}$ and WT controls (Fig S5). In contrast with these findings, there were no obvious changes in the size and number of OC in bone tissues from SHIP1-deficient versus control strains (Fig S6).

The involvement of SHIP1 in in vitro OPC proliferation was further substantiated by inhibition of the enzyme using the SHIP1-specific inhibitor 3α-aminocholestane (3AC). For this purpose, OPC from WT, $SHIP1^{+/styx}$, and $SHIP1^{styx/styx}$ mice were cultured in media supplemented with CSF-1, RANKL, and varying concentrations of 3AC. Treatment of the cultures with 3AC led to an increase of 55% in the number of cells in cultures of WT *and* $SHIP1^{+/styx}$ OPC at a concentration of 2.5 nM, whereas at 5 nM, 3AC caused cell death in the cultures of WT *and* $SHIP1^{+/styx}$ OPC but not in cultures of $SHIP1^{styx/styx}$ OPC (Fig 2G). These data demonstrate that both genetic and pharmaceutical blockade of SHIP1 promote OPC proliferation in vitro.

### SHIP1 deficiency reduces the capacity of OC to dissolve amorphous calcium phosphate

The capacity of OC to dissolve amorphous calcium phosphate (CaP) was investigated using functional tests. For this purpose, the cells were seeded onto a layer of CaP spiked with $^{45}$Ca in media supplemented with varying concentrations of RANKL. Whereas the mineral dissolution capacities of OC derived from WT and $SHIP1^{+/styx}$ OPC did not differ, a deficiency in SHIP1 led to a 75% reduction in OC activity in comparison with WT (Fig 2H). In cultures of OPC derived from $Rag2^{-/-}/Il2rg^{-/-}/SHIP1^{styx/styx}$ mice, SHIP1 deficiency led to an overall 35% reduction in mineral dissolution activity. Furthermore, OC derived from OPC of $Rag2^{-/-}/Il2rg^{-/-}/SHIP1^{+/styx}$ immunodeficient animals showed reduced CaP-dissolution activity (Fig 2I). In summary, SHIP1 modulates OPC proliferation and the activity of mature OC.

### SHIP1 regulates the OC-associated transcriptomic signature

To identify SHIP1-dependent regulation of gene expression in OC, transcriptome analysis was performed on OPC grown in cultures supplemented with or without RANKL. Principal component analysis (PCA) was performed on the top 1,000 genes with the most variable expression in all samples. The first component (83% variance) discriminated between treatments with or without RANKL, and the second component (7% variance) distinguished between cells with $Rag2^{-/-}/Il2rg^{-/-}$ and $Rag2^{+/+}/Il2rg^{+/+}$ backgrounds (Fig S7A). The PCA between RANKL-treated groups established a distinct segregation of $SHIP1^{styx/styx}$ samples in comparison with the control groups only in the $Rag2^{+/+}/Il2rg^{+/+}$ background (with 40% variance in the first component) (Fig S7B and C).

Differential expression analysis was performed to define a list of OC-associated transcripts. Levels of more than 600 transcripts were up-regulated (FDR-corrected $P$-value < 0.05, $\log_2$ fold change > 2) in OC derived from WT OPC on day 5 of culture. The expression levels of the 50 most up-regulated transcripts in this list were assessed in the other experimental groups. In line with the results of PCA, we observed clustering of $Rag2^{+/+}/Il2rg^{+/+}$ versus $Rag2^{-/-}/Il2rg^{-/-}$ OC. In addition, $SHIP1^{styx/styx}$ cells were grouped (Fig 3). These results were further confirmed by qRT-PCR for OC markers, specifically *Calcr* and *Ctsk* (Fig S8A–D). Next, transcriptome analysis was performed on OC cultures, which revealed 362 genes that were differentially expressed between $SHIP1^{styx/styx}$ and WT groups, and 460 genes that were differentially expressed between the $Rag2^{-/-}/Il2rg^{-/-}/SHIP1^{+/+}$ and $Rag2^{-/-}/Il2rg^{-/-}/SHIP1^{styx/styx}$ groups (FDR-corrected $P$-value < 0.05; Fig 4A and B). Among these two sets of genes, 118 genes were found in both groups (Fig 4C).

Further analyses of down-regulated genes in the $SHIP1^{styx/styx}$ OC indicated enrichment of three gene ontology (GO) terms: bone remodelling, bone resorption, and OC differentiation, whereas the up-regulated genes were enriched mostly in cell–cell adhesion and migration processes (Fig S9A and B). KEGG pathway analysis revealed up-regulation of pathways related to protein synthesis, such as ribosome biogenesis and biosynthesis of amino acids, PPAR, and PI3K/AKT signalling pathways (Fig S9C). The same analyses performed on $Rag2^{-/-}/Il2rg^{-/-}$ cells highlighted differences in the GO terms cell fusion and metabolism, as well as cytokine signalling and metabolic pathways in the KEGG ontology (Fig S9D, E, and F). Overall, these data demonstrate changes in the transcriptome signature of SHIP1-deficient OC, with down-regulation of the transcripts controlling OC development.

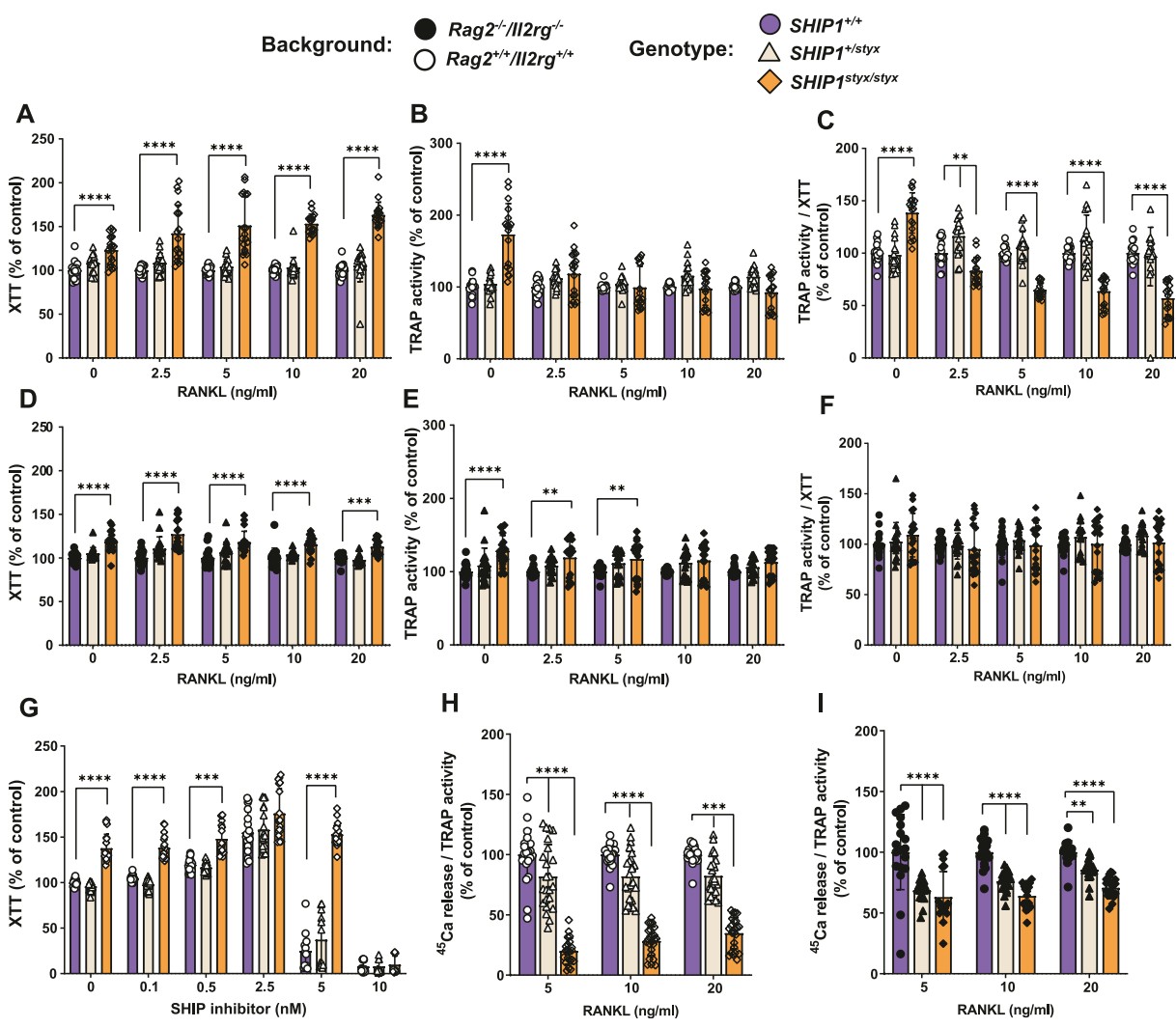

**Figure 2. Cell proliferation is accelerated in osteoclast progenitor cells (OPC) from SHIP1-deficient mice.**
**(A, B, C, D, E, F)** OPC from WT; *SHIP1^+/styx^; SHIP1^styx/styx^* (A, B, C) and *Rag2^−/−^/Il2rg^−/−^/SHIP1^+/+^*; *Rag2^−/−^/Il2rg^−/−^/SHIP1^+/styx^*, and *Rag2^−/−^/Il2rg^−/−^/SHIP1^styx/styx^* (D, E, F) mice were cultured in media supplemented with CSF-1 and RANKL. **(A, B, C, D, E, F)** After 5 d in culture, viable cells (A, D) and TRAP activity levels (B, E) were determined and TRAP activity/XTT (C, F) was calculated. **(A, C)** After 5 d in culture, viable cells in cultures of OPC from *SHIP1^styx/styx^* mice were higher than those in WT and *SHIP1^+/styx^* mice (A), whereas TRAP activity/XTT was lower except when the cells were grown without RANKL (C). **(D, F)** In cultures of OPC from *Rag2^−/−^/Il2rg^−/−^* mice, the increase in viable cells was attenuated (D), as was TRAP activity/XTT (F). **(G)** OPC from WT, *SHIP1^+/styx^, SHIP1^styx/styx^* mice were cultured in media supplemented with CSF-1, RANKL, and the SHIP1 inhibitor 3AC at the indicated concentrations for 5 d. At 3AC concentrations of 0.1 and 0.5 nM, no effects of SHIP1 inhibition could be detected. At a 3AC concentration of 2.5 nM, the difference in the proliferation rate between WT, *SHIP1^+/styx^*, and *SHIP1^styx/styx^* OPC was blunted. At 5 nM 3AC, the number of viable cells decreased in the cultures of OPC from WT and *SHIP1^+/styx^* mice, but not in cultures of OPC from *SHIP1^styx/styx^* mice. At 10 nM 3AC, no cell viability could be detected. **(H, I)** Mineral dissolution activity of osteoclasts (OC) in vitro. OPC from (H) WT, *SHIP1^+/styx^*, and *SHIP1^styx/styx^* and (I) *Rag2^−/−^/Il2rg^−/−^/SHIP1^+/+^*; *Rag2^−/−^/Il2rg^−/−^/SHIP1^+/styx^*, and *Rag2^−/−^/Il2rg^−/−^/SHIP1^styx/styx^* mice were cultured for 5 d in media supplemented with CSF-1 and RANKL. Subsequently, the OC were transferred onto a layer of amorphous CaP spiked with ^45^Ca and cultured for another 24 h with the indicated concentrations of RANKL. **(H)** OC derived from *SHIP1^styx/styx^* OPC showed 70% lower CaP-dissolution capacity than controls. **(I)** Additional deficiency in IL2RG expression partially reversed this effect. Data indicate mean ± SD of three independent experiments, each with one mouse per group as a cell donor. Significant changes were calculated with two-way ANOVA, **($P < 0.01$), ***($P < 0.001$), and ****($P < 0.0001$).

## Inhibition of mTOR using rapamycin

Since the transcriptome data indicated the up-regulation of biological processes involved in protein synthesis in cells from SHIP1-deficient animals, the possible role of mTOR complex 1 (mTORC1) in the activation of AKT was investigated. WT and *SHIP1^styx/styx^* OPC were cultured in media supplemented with RANKL (20 ng/ml), CSF-1 (30 ng/ml), and varying concentrations of the mTORC1 inhibitor rapamycin. TRAP/XTT was evaluated on day 5 of culture. The data demonstrate that rapamycin in WT OPC increased TRAP activity/XTT by 35% and 45% at concentrations of 0.05 and 0.25 nM, respectively, whereas the addition of higher doses of rapamycin exerted an inhibitory effect ($P < 0.0001$) (Fig 5A). In cultures of *SHIP1^styx/styx^* OPC, treatment with rapamycin resulted in a more modest stimulatory effect on TRAP activity/XTT, with a 30% increase at 1.25 nM, whereas higher doses led to an inhibitory effect ($P < 0.0001$; Fig 5A). These data show that TORC1 is not critically involved in the OC phenotype caused by SHIP1 deficiency.

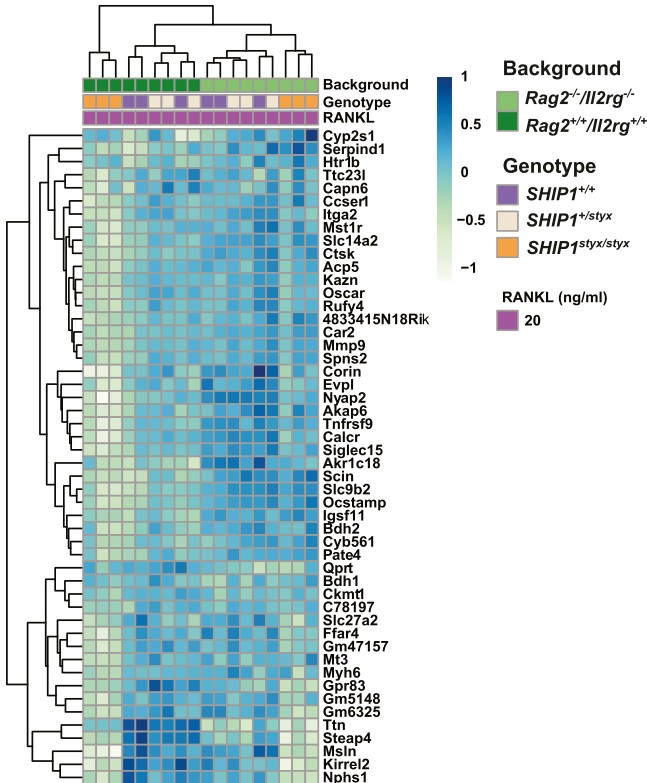

**Figure 3. Heat map of the selected top 50 genes in OPC cultured in media supplemented with 20 ng/ml RANKL.**

$Rag2^{+/+}/Il2rg^{+/+}/SHIP1^{styx/styx}$ OC demonstrated down-regulation in most of the OC-specific markers. Statistical significance was defined by an adjusted $P$-value< 0.05, $\log_2$ fold change > 2. Data are from three mice per group.

### Modulation of OPC proliferation by IL-15

Mature OC were found to express $Il2rg$ transcripts (Fig S3B); therefore, a possible contribution of this gene product to the SHIP1-deficient OC phenotype was addressed. Among the cytokines encoded by transcripts detected in in vitro generated OC (IL1$\beta$, IL10, IL15, IL16, and IL18), only IL-15 requires IL2RG (the common-gamma chain receptor) for signalling (Fig S10).

To evaluate the effects of IL-15 on OPC proliferation, the culture media were supplemented with CSF-1, RANKL, and IL-15. At lower levels of CSF-1 (15 ng/ml), 100 ng/ml IL-15 enhanced proliferation by 40% in WT and 67% in $SHIP1^{styx/styx}$ OPC. At higher concentrations of CSF-1 (30 ng/ml), the same levels of IL-15 enhanced OPC proliferation by 30% in WT OPC and 33% in $SHIP1^{styx/styx}$ OPC (Fig 5B). Taken together, these data demonstrate the stimulatory effect of IL-15 on OPC proliferation, which was attenuated by increasing concentrations of CSF-1.

## Discussion

Numerous skeletal diseases are associated with or driven by local and systemic inflammatory processes. This interdependence of bone and the immune system is not limited to the close anatomical proximity of these structures, but is also based on a multitude of cellular and biochemical functions involving their cellular components during health and disease (31). Research on this topic has been merged in the field of osteoimmunology. Changes in skeletal metabolism associated with chronic inflammatory disorders remain clinically relevant, highlighting the need to further evaluate the central signalling pathways and molecular effectors for therapy.

SHIP1-deficient mice represent a paradigm of chronic inflammation with severe consequences for the skeletal system. Indeed, SHIP1 deficiency provokes a chronic inflammatory disease characterised by high systemic levels of multiple pro-inflammatory mediators, which are associated with low body weight, delayed skeletal development, and general wasting syndrome. In the present study, we established that the genetic abrogation of lymphoid cells, accompanied by a partial reduction in the serum levels of inflammatory cytokines, corrects this phenotype by normalising skeletal growth and body mass in SHIP1-deficient animals. Therefore, this model enables the uncoupling of SHIP1-dependent systemic inflammation mediated by lymphoid cells from the cell-autonomous effects of SHIP1 on bone cell lineages.

Our data demonstrating skeletal alterations in SHIP1-deficient $SHIP1^{styx/styx}$ animals (27) are consistent with those of previous studies showing that $SHIP1^{-/-}$ mice display a low bone mass phenotype (28). However, in these previous reports, the question of whether the reduction in skeletal development was caused by delayed growth of these animals due to systemic inflammation or whether it was produced by an intrinsic effect of SHIP1 deficiency on bone cell lineages was not addressed. To answer this conundrum, the microarchitecture of the femora and vertebrae from SHIP1-deficient mice was compared with the bone structure in WT mice from birth to 12 wk of age. Although bone mass and density (BV/TV) were lower in SHIP1-deficient mice than in age-matched WT controls, the observed differences correlated with the bone mass and microarchitecture in size-matched WT control animals. Thus, the reported low bone mass in mice lacking SHIP1 is likely a consequence of impaired growth rather than SHIP1-dependent changes in bone development and turnover. This notion is further supported by studies showing that selective deletion of SHIP1 in OC or OB cell lineages failed to induce bone loss (29), demonstrating that SHIP1 deficiency in bone cell lineages had no significant impact on skeletal development.

Because immunodeficient $Rag2^{-/-}/Il2rg^{-/-}$ mice in the present study did not exhibit any skeletal phenotype, we infer that systemic inflammatory processes are responsible for the observed decrease in body weight and skeletal growth associated with SHIP1 deficiency. Thus, the lymphoid lineage is essential for phenotypic characterisation of SHIP1 KO strains. However, conditional ablation of SHIP1 in B/T or myeloid cells was not associated with the development of traits accompanying global SHIP1 deficiency. These findings indicate the overlapping contributions of several immune cell types to SHIP1-regulated inflammatory diseases and bone alterations.

SHIP1 can regulate the development of OC lineage cells, with a consequent increase in the number of OC in bones from SHIP1-deficient animals (32). In the present study, transcriptome analysis of bone tissues from SHIP1-deficient animals revealed higher expression levels of OC gene markers in these bones than in the WT control bones. To elucidate whether SHIP1 exerts a cell-autonomous effect

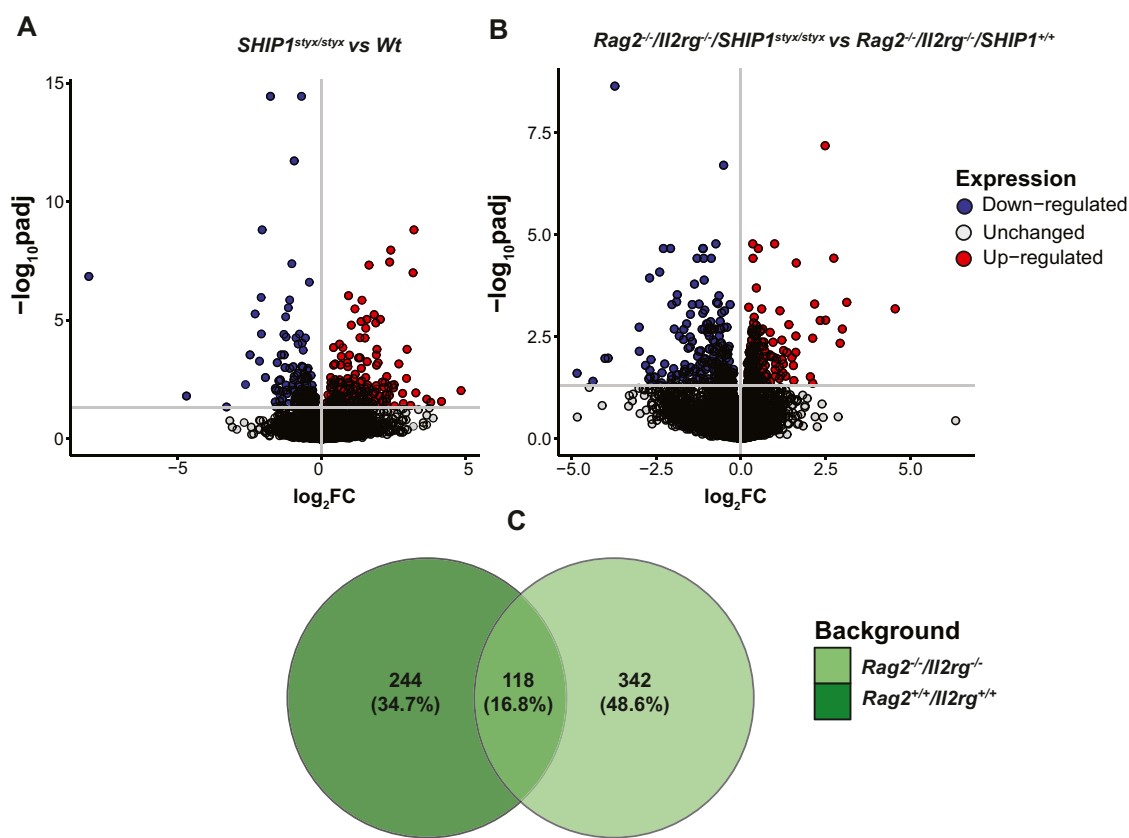

**Figure 4.  Differentially expressed transcripts in SHIP1-deficient OC in comparison with controls.**
**(A, B)** Volcano plot representing the number of differentially expressed genes in *SHIP1^{styx/styx}* versus WT OC and *Rag2^{−/−}/Il2rg^{−/−}/SHIP1^{styx/styx}* versus *Rag2^{−/−}/Il2rg^{−/−}/SHIP1^{+/+}* OC, respectively. **(C)** A total of 118 genes were differentially expressed in both *Rag2^{+/+}/Il2rg^{+/+}* and *Rag2^{−/−}/Il2rg^{−/−}* OC. Statistical significance was defined as adjusted *P*-value < 0.05, log_2 fold change > or < 0. Data are from three mice per group.

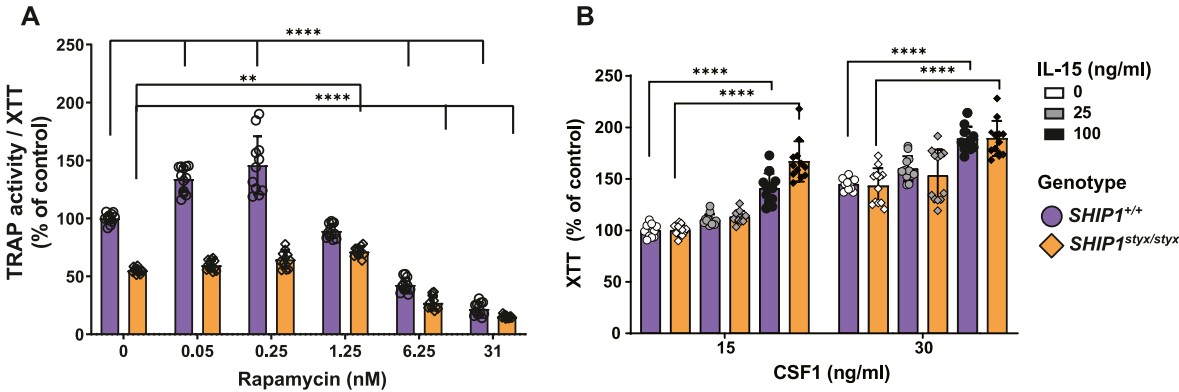

**Figure 5.  Role of mTORC1 and IL-15.**
**(A)** OPC from WT *and SHIP1^{styx/styx}* animals were cultured with different concentrations of rapamycin, 20 ng/ml of RANKL, and 30 ng/ml of CSF-1. The development of OC was determined by calculation of TRAP activity/XTT. In WT OPC, rapamycin enhanced TRAP activity/XTT at low doses and showed an inhibitory effect at higher doses. In *SHIP1^{styx/styx}* OPC, rapamycin showed a stimulatory effect on TRAP activity/XTT only at 1.25 nM, and higher doses led to inhibition. **(B)** OPC from WT *and SHIP1^{styx/styx}* animals were cultured with different concentrations of IL-15, 20 ng/ml of RANKL, and different concentration of CSF-1. The proliferation rates were detected using the XTT assay. In the presence of 15 and 30 ng/ml CSF-1, IL-15 at a concentration of 100 ng/ml induced the proliferation of WT and *SHIP1^{styx/styx}* OPC. Data indicate mean ± SD of two independent experiments, each with one mouse per group as a cell donor. Significant changes were calculated with two-way ANOVA, **(*P* < 0.01), ***(*P* < 0.001), and ****(*P* < 0.0001).

on OC lineage cells or whether this gene expression pattern is modulated by the bone microenvironment, OC development was assessed in vitro. Our data demonstrate that SHIP1 deletion does affect OC development. Consistent with the published findings (28, 33), we found that SHIP1 deficiency in OPC enhanced their proliferation. Furthermore, the SHIP1-specific inhibitor 3AC induced higher proliferation of WT OPC, confirming that SHIP1 controls cell proliferation in vitro. Because the applied cell culture system of OPC was devoid of a bone microenvironment, we conclude that SHIP1 modulates OC lineage development in a cell-intrinsic manner. Moreover, the overexpression of OC traits in the bones of SHIP1-deficient animals suggests an additional regulatory contribution of the bone microenvironment to OC development in these mice.

In contrast to the findings of a previous report (28), in the present study, the capacity of SHIP1-deficient OPC to develop into mature OC was reduced and the mineral dissolution activity of OC in culture was decreased in comparison with WT control OPC. Although there is no obvious explanation for this discrepancy, the different results may be rooted in the different natures of the employed OPC. Herein, 24-h-cultured CSF-1-dependent non-adherent cells were used as OPC, whereas in a previous study, bone marrow-derived macrophages cultured for 6 d with CSF-1 were assayed. Given the distinct proliferation rates of SHIP1-deficient and -proficient OPC, this difference may lead to changes in the composition of OPC populations. In the present study, the decrease in OC development from OPC derived from SHIP1-deficient bone marrow was further corroborated by transcriptome analysis, which revealed reduced expression of OC traits in cultures derived from SHIP1-deficient versus WT OPC. However, in both studies, an increase in the size of the OC with a large number of nuclei was observed.

As mentioned previously, the wasting syndrome and skeletal phenotype associated with SHIP1 deficiency were ameliorated in immunodeficient mice lacking RAG2 and IL2RG. Therefore, we evaluated whether osteoclastogenesis in OPC lacking RAG2, IL2RG, or SHIP1 differed from that of SHIP1-deficient OPC. Consistent with SHIP1-deficient OPC, triple-deficient OPC showed an increase in proliferation rate and unchanged OC development capacity in comparison with WT or $Rag2^{-/-}/Il2rg^{-/-}$ controls. However, deficiencies in RAG2 and IL2RG could partially reverse the diminished mineral dissolution activity of SHIP1-deficient OC, suggesting that a lack of either RAG2, IL2RG, or both may partially compensate for SHIP1 deficiency via a cell-autonomous mechanism, prompting us to investigate the role of these genes in this model.

Although $Rag2$ transcripts were undetectable in OC, these cells expressed $Il2rg$ gene products that encode a common subunit of multiple interleukin receptors. Our RNA sequencing data suggest that among these interleukins, only IL-15 is expressed by OC, highlighting the possibility of autocrine/paracrine stimulation of OC development by this cytokine. Indeed, IL-15 increases the number of OPC during the early stages of OC development (34). In the present study, IL-15 was found to induce in vitro the proliferation of WT OPC and, to a greater extent, SHIP1-deficient OPC. Thus, IL-15 signalling increases, in synergy with CSF-1 signalling, the proliferation rate of OPC through the PI3K/AKT pathway. Therefore, the decreased proliferation of $Rag2^{-/-}/Il2rg^{-/-}/SHIP1^{styx/styx}$ OPC in comparison with $SHIP1^{styx/styx}$ cells stems from their inability to engage in IL-15 signalling.

Transcriptome data indicated the up-regulation of protein synthesis in OC lacking SHIP1. On the basis of previous reports suggesting impaired OC differentiation upon activation of PI3K/AKT/mTOR signalling in OPC (35), we tested whether the stimulation of protein synthesis was caused by the activation of mTORC1 in the absence of SHIP1. This assumption was supported by the observation in the present study showing that low doses of the mTORC1 inhibitor rapamycin caused an increase in OC development in WT OPC. A relevant role of mTORC1 in SHIP1-modulated OC development, however, was ruled out when mTORC1 blockade failed to restore OC development from SHIP1-deficient OPC compared with WT OPC.

In conclusion, the results of our study provide further evidence for the contribution of immune cells to the development of wasting diseases and retardation of body and bone growth in SHIP1-deficient mice. Impairment of skeletal growth in SHIP1 KO animals is a secondary phenotype subsequent to the inflammatory condition. Depletion of lymphoid cell populations attenuates the whole-body and skeletal phenotypes caused by SHIP1 deficiency. These data also demonstrate differences in the in vivo and in vitro OC phenotypes caused by SHIP1 deficiency. The lack of SHIP1 leads to a reduction in OC development, which cell-autonomously affects OPC, whereas OC development may be enhanced in vivo because of an osteoclastogenic inflammatory microenvironment in the bones of SHIP1 KO animals. Restoration of the skeletal development in $Rag2^{-/-}/Il2rg^{-/-}/SHIP1^{styx/styx}$ in comparison with $SHIP1^{styx/styx}$ animals demonstrates that the cell-intrinsic effects of SHIP1 deficiency in OC are blunted under physiological conditions. However, it remains to be elucidated whether SHIP1 deficiency leads to more pronounced changes in skeletal development and homeostasis in pathological conditions such as post-menopausal osteoporosis with underlying chronic low-level inflammation.

# Materials and Methods

### Animals

All animal strains were maintained on a $C57BL6/J$ background. The $SHIP1^{styx/styx}$ ($Inpp5d^{m1Btlr/m1Btlr}$, http://www.informatics.jax.org/allele/key/65037) strain has been previously described (27). Homozygous mice of this strain lack SHIP1 protein expression and display the same phenotype as $Inpp5d$ KO strains (26, 36). $Rag2^{-/-}/Il2rg^{-/-}/SHIP1^{styx/styx}$ were generated by crossing $SHIP1^{styx/styx}$ mice with $Rag2^{tm1Fwa}$ (37) and $Il2rg^{tm1Wjl}$ mice (38). All animals were maintained under specific opportunistic pathogen-free conditions at the Central Animal Facility of the University of Bern, Switzerland. For ex vivo and in vitro experiments, mice were age- and sex matched.

### Quantification of serum cytokines

The serum cytokine levels were measured using multiplexing laser beads (Eve Technologies).

### OC development in vitro

Bone marrow cells were isolated from femora, tibiae, and humeri of 8–12-wk-old mice. Bone marrow cells were centrifuged at 250$g$ for

7 min at 4°C and distributed into two 75-cm² cell culture flasks in α-MEM (GIBCO, LuBio Science) supplemented with 10% heat inactivated FBS, 1% penicillin G/streptomycin (Pen/Strep, GIBCO, LuBio Science), and 30 ng/ml CSF-1 (FDP-Chiron). After 24 h, non-adherent CSF-1–dependent OPC were re-plated in multiwell plates at a density of $2 \times 10^5$ cells/ml in α-MEM supplemented with 10% FBS. To induce OC formation, the medium for OPC was supplemented with CSF-1 (30 ng/ml) and RANKL (20 ng/ml; PeproTech, LuBio Science). For SHIP1 inhibition, culture media were supplemented with the inhibitor 3α-aminocholestane (3AC). The medium was changed after 3 d of culture, and the cells were collected after 4, 5, and 6 d for further assessment.

### Cell viability (XTT) assay

OPC were seeded into 96-well plates and cell viability was assessed using a Cell Proliferation Kit (XTT, Roche Diagnostics) according to the manufacturer's instructions. Briefly, at day 5 of the culture, cells were incubated with the XTT labelling mixture for 4 h at 37°C. Subsequently, absorbance was measured at 470 nm using a SPARK MULTIMODE spectrophotometer (Tecan Group Ltd). The reference wavelength was set at 630 nm.

### Determination of OC differentiation

Tartrate-resistant acid phosphatase (TRAP) activity was quantified on day 5 of culture. Briefly, cells were rinsed once with PBS (137 mM NaCl/2.7 mM KCl/12 mM Pi, pH 7.4), lysed in 0.1% Triton X-100/1 M NaCl, and frozen at –20°C. After one freezing cycle, 4.61 mg/ml 4-nitrophenyl phosphate (Sigma-Aldrich) in 40 mM Na-tartrate/50 mM Na-acetate (pH 4.8) was added to the cell lysate. The enzyme reaction was stopped with 0.2 M NaOH after 1 h, and absorbance was measured at 405 nm with a SPARK MULTIMODE spectrophotometer using 690 nm as the reference wavelength.

To visualise OC, cells were stained for TRAP using a commercial kit (Sigma-Aldrich) in accordance with the manufacturer's protocol. Briefly, the cells were washed with PBS and fixed with 4% PFA (Merck) for 10 min at RT. After rinsing three times with demineralised water, the cells were incubated with the TRAP staining solution for 5 min. The stained cultures were rinsed three times, air-dried, and analysed.

### Mineral dissolution assay

To quantify the capacity of in vitro generated OC to dissolve amorphous calcium phosphate (CaP), mature OC were harvested and re-plated onto a layer of amorphous CaP spiked with $^{45}$Ca, as described previously (39). For this purpose, 0.12 M Na$_2$HPO$_4$ and 0.2 M CaCl$_2$ (50 mM Tris–HCl, pH 7.4) containing $^{45}$Ca (Perkin Elmer) at a concentration of 200,000 Bq/ml were preincubated overnight in 5% CO$_2$ at 37°C. After mixing equal volumes of these solutions, CaP precipitated as a slurry. The CaP slurry was rinsed twice with sterile distilled water and resuspended in distilled water at a concentration of 50 μl/ml. Two hundred microliters/well of the suspension (containing 4,000 Bq $^{45}$Ca) were transferred into 24-well plates and then dried under sterile conditions at RT for 3 d followed by a heating step of 80°C for 3 h. Before seeding the cells onto CaP, the

plates were incubated with α-MEM supplemented with 30% FBS overnight in 5% CO$_2$ at 37°C.

OC were generated by growing OPC in 20-cm² dishes in medium supplemented with 30 ng/ml CSF-1 and 20 ng/ml RANKL for 5 d, as described above. To release the cells from the culture dish, OC were incubated with 0.02% EDTA in PBS (Merck, Sigma-Aldrich) at 37°C. After 15 min, the detached cells were centrifuged at 200$g$ for 5 min and resuspended in a 500 μl/20 cm² dish. Fifty microliters of the cell suspension were transferred into CaP-coated 48-well plates, and 200 μl of α-MEM and 10% FBS supplemented with HCl (15 mM), CSF-1 (30 ng/ml), RANKL (5, 10, and 20 ng/ml) was added. After 24 h, $^{45}$Ca released into the supernatant was measured using a scintillation counter. The cells were subsequently lysed to measure the total TRAP activity. The amount of radioactivity in the culture medium, normalised to the TRAP activity, was used as an indicator of the mineral-dissolution capacity of the OC.

### X-ray and micro-computed tomography

The femora and vertebrae were excised, fixed in 4% PFA in PBS for 24 h, washed with running tap water for 4 h, and transferred to 70% ethanol. X-ray imaging was performed using the TrueView 100 Pro high-resolution X-ray machine (CompAI).

Micro-CT analysis was performed using a micro-CT (MicroCT40; SCANCO Medical AG). The samples were placed along the longitudinal axis perpendicular to the X-ray beam. The X-ray source was set at 70 kVp and 114 mA with an integration time of 300 ms. Measurements were performed using built-in software from Scanco (Scanco Module 64-bit; V5.15). Regions of interest were manually delineated in the vertebral bodies of the L4 and distal femora. The measurements were performed at the highest resolution with a voxel size of 6 μm.

### Analysis of gene expression by real-time quantitative reverse transcription PCR

To assess gene expression, OPC were cultured in six-well plates for 5 d. Total RNA was extracted using NucleoSpin RNA Plus Kit (Macherey-Nagel) and was reverse-transcribed using M-MLV reverse transcriptase (200 U) (Promega), random primers (50 μg/ml) (Promega), the nucleotide mix (500 μM) (Roche, Sigma-Aldrich), and RNase inhibitor (25 U) (Roche). Subsequently, TaqMan RT–PCR (qRT-PCR) was performed on a Quantstudio 7 System (Life Technologies/Applied Biosystems) using TaqMan Fast Universal PCR Master Mix (Thermo Fisher Scientific, Life Technologies Europe BV) and Assays-on-Demand (Thermo Fisher Scientific, Life Technologies Europe BV) for *Inpp5d* (Mm00494987_m1), *Rag2* (2033272 A6), *Il2rg* (Mm00442885_m1), *Ctsk* (Mm00484039_m1), and *Calcr* (Mm00432282_m1). The reaction mixtures were denatured for 20 s at 95°C, followed by 45 amplification cycles of 3 s at 95°C and 30 s at 60°C. The expression of target genes was normalised against the levels of the housekeeping gene β-glucuronidase (*Gusb*; Mm01197698_m1) mRNA.

### Histology

Vertebrae and tibiae were excised and fixed for 24 h in 4% PFA in PBS. After decalcification and dehydration, tissues were embedded in

paraffin and 5-µm sections were prepared. Sections were stained for TRAP, as described above, and counterstained with hematoxylin. Slides were scanned by staff of the Translational Research Unit of the Institute of Tissue Medicine and Pathology using a whole slide Pannoramic 250 Flash (3DHISTECH) scanner.

### Bulk RNA sequencing

For RNA extraction and library preparation, the diaphysis of one femur (n = 3 for each genotype) was dissected and placed in RNALater (Sigma-Aldrich). Total RNA was extracted by homogenising the bone samples in microtubes containing metal beads and 1 ml of TRIzol (Invitrogen) using a tissue homogeniser (TissueLyser 2; QIAGEN), followed by digestion with DNase using the NucleoSpin RNA Plus Kit (Macherey-Nagel) according to the manufacturer's instructions.

For cell culture samples, OPC (for each genotype, n = 3) were cultured in six-well plates with (20 ng/ml) or without RANKL for 5 d. Total RNA was extracted using the NucleoSpin RNA Plus Kit (Macherey-Nagel) according to the manufacturer's instructions. The quantity and quality of RNA samples were assessed using a Qubit 2.0 Fluorometer (Thermo Fisher Scientific). Only RNA samples with an RNA integrity number (RIN) above eight were considered for RNA-Seq. A cDNA library was prepared for each RNA sample using the TruSeq Stranded mRNA Sample Preparation Kit (v2; Illumina). Libraries from all samples (n = 36) were sequenced using the next-generation sequencing (NGS) Platform of the University of Bern using an Illumina NovaSeq 6000 with 50-bp paired-end reads at a minimum of 18 million reads per sample.

### RNA-sequencing analysis

Read-quality assessment was performed using FastQC version 0.11.5 (http://www.bioinformatics.babraham.ac.uk/projects/fastqc). Reads were mapped to the *Mus musculus* reference genome Ensembl GRCm38.p6 using Hisat2 version 2.1.0 (40). The number of reads aligned to the exons was counted using featureCounts version 1.6.0 (41). Differential gene expression analysis was performed with R version 4.1.2 (https://www.R-project.org/) using the DESeq2 package version. 1.34.0 (42). The following comparisons were used to identify differentially expressed genes (DEG) in tissue samples: bone from each genotype ($SHIP1^{+/styx}$ and $SHIP1^{styx/styx}$) versus control ($SHIP1^{+/+}$), and for cell culture samples: OPC treated with 20 ng/ml RANKL versus cells treated without RANKL in WT OPC and OPC from each genotype versus the WT controls. Genes with an FDR-corrected *P*-value less than 0.05 were considered to be differentially expressed. Heat maps were generated using the pheatmap R package (https://CRAN.R-project.org/package=pheatmap). Gene ontology (GO) enrichment and KEGG pathway analysis were performed using the enrichGO and gseKEGG functions, respectively, of the clusterProfiler R package version 4.2.2 (43) along with the Bioconductor package org.Mm.eg.db containing the genome-wide annotation for mouse.

### Statistics

The results are expressed as mean ± SD from individual experiments. Statistical analyses were performed using GraphPad Prism 8.4.1 for Windows (GraphPad Software). As mentioned in the figure legends, either one-way analysis of variance (ANOVA) or two-way ANOVA with Dunnett's multiple comparison test was used to detect statistically significant data. Differences between experimental groups were considered to be statistically significant only when $P < 0.01$. Unless otherwise indicated, all experiments were performed independently at least three times.

### Study approval

Animal experiments were performed in accordance with Swiss Federal regulations and approved by the Bernese Cantonal Veterinary Office (permit numbers BE20/18 to P Krebs and BE22/22 to W Hofstetter).

## Data Availability

RNA sequencing data form femoral diaphysis and primary OPC cultures from $SHIP^{styx/styx}$, $SHIP1^{+/styx}$, $WT$ (C57BL6/J), $Rag2^{-/-}/Il2rg^{-/-}/SHIP1^{styx/styx}$, $Rag2^{-/-}/Il2rg^{-/-}/SHIP1^{+/styx}$, and $Rag2^{-/-}/Il2rg^{-/-}/SHIP1^{+/+}$ mice (n = 3 for each genotype) can be downloaded from the ArrayExpress Archive of Functional Genomics Data, where they are stored under the title "Transcriptomic analysis of femoral diaphysis and primary OPC cultures from mouse strains with SHIP1 deficiency and controls" (Reference number E-MTAB-13164).

## Supplementary Information

## Acknowledgements

The authors thank Margaux Bringardner and Franziska Strunz for their help and support during this project. This study was partly supported by grants from the Swiss National Science Foundation (310030_138188 and 314730_163086), the Bern University Research Foundation, the Swiss Life Anniversary Foundation, as well as by the Research Fund of the Swiss Lung Association, Bern, and the Bernese Lung League (Lungenliga Bern) (all of them to P Krebs) and the Robert Mathys Foundation, Bettlach (to W Hofstetter).

### Author Contributions

F Safari: data curation, formal analysis, validation, investigation, visualization, methodology, and writing—original draft.
WJ Yeoh: data curation, formal analysis, validation, investigation, and methodology.
S Perret-Gentil: formal analysis and investigation.
F Klenke: formal analysis, validation, and investigation.
S Dolder: formal analysis and investigation.
W Hofstetter: conceptualization, resources, supervision, funding acquisition, validation, investigation, visualization, methodology, project administration, and writing—original draft, review, and editing.
P Krebs: conceptualization, resources, supervision, funding acquisition, validation, investigation, visualization, methodology,

project administration, and writing—original draft, review, and editing.

## Conflict of Interest Statement

The authors declare that they have no conflict of interest.

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
