## [Reviewer comments · Life Science Alliance]

SHIP1 deficiency causes inflammation-dependent retardation in skeletal growth

Fatemeh Safari, Wen Yeoh, Saskia Perret-Gentil, Frank Klenke, Silvia Dolder, Willy Hofstetter, and Philippe Krebs
DOI: <https://doi.org/10.26508/lsa.202302297>

Corresponding author(s): Philippe Krebs, Institute of Tissue Medicine and Pathology, University of Bern and Willy Hofstetter, University of Bern

Review Timeline:	Submission Date:	2023-07-31
	Editorial Decision:	2023-09-06
	Revision Received:	2023-12-04
	Editorial Decision:	2024-01-09
	Revision Received:	2024-02-05
	Accepted:	2024-02-06

Transaction Report:

September 6, 2023

Re: Life Science Alliance manuscript #LSA-2023-02297-T

Prof. Philippe Krebs
Institute of Tissue Medicine and Pathology, University of Bern
Murtenstrasse 31
Bern CH - 3008
Switzerland

Dear Dr. Krebs,

Thank you for submitting your manuscript entitled "SHIP1 deficiency causes inflammation-dependent retardation in skeletal growth" to Life Science Alliance. The manuscript was assessed by expert reviewers, whose comments are appended to this letter. We invite you to submit a revised manuscript addressing the Reviewer comments.

Thank you for this interesting contribution to Life Science Alliance. We are looking forward to receiving your revised manuscript.

Sincerely,

B. MANUSCRIPT ORGANIZATION AND FORMATTING:

Reviewer #1 (Comments to the Authors (Required)):

Life Science Alliance Review "SHIP1 deficiency causes inflammation-dependent retardation in skeletal growth"

This manuscript by Safari et al explores the skeletal and osteoclast-specific effects of immunodeficiency in global SHIP1-deficient mice. Because global SHIP1 KO mice have whole body inflammation and are runted in size, the authors crossed the SHIP1 KO mice to immunodeficient RAG2-^{-/-}IL2R β -^{-/-} mice to remove systemic inflammation. They show that Rag2^{-/-}/Il2rg^{-/-}/SHIP1^{styx/styx} mice have normalized bone growth and bone mass compared to SHIP1^{styx/styx} mice. Bone marrow s and osteoclast transcriptomic was performed and shows significant changes in SHIP1 KO with and without lymphocytes. Analysis of osteoclastogenesis ex vivo shows increased proliferation of SHIP1^{styx/styx} preosteoclasts independently from lymphocytes. Trap activity of SHIP1^{styx/styx} osteoclasts was reduced but normalized in the absence of lymphocytes. Use of 3AC SHIP1 inhibitor increased proliferation of WT cells. They show that rapamycin promotes WT osteoclast differentiation but not SHIP1-deficient preosteoclasts. Lastly, they show that IL15 synergistically promotes preosteoclast proliferation with mCSF but this is independent of SHIP1 expression.

Overall, this is an interesting paper with some novel aspects (Use of immunodeficient mice with SHIP1 KO, transcriptional data, finding of IL15 promoting proliferation).

The conclusions are mostly supported by the data. However, discussion is lacking in terms of putting these results in the context of mesenchymal stem cell-specific and myeloid-specific SHIP1 KO mice published by Iyer et. all in 2014. Prior studies by Iyer et. all, showed that mice with myeloid cell specific deletion of SHIP1 in osteoclast precursors had normal bone mass and osteoclasts in vivo but substantially larger TRAP⁺ osteoclasts in vitro. They also showed that mice with mesenchymal stem cell-specific (OSXCre) depletion of SHIP1 have runted size, low bone mineral apposition and low bone mass in the absence of inflammation. Importantly, they showed that MSC-SHIP1 deficient mice also had lower osteoclast numbers in vivo and in vitro. They conclude that SHIP1 expression in the mesenchymal stem cell compartment provides a negative regulation of osteoclastogenesis in vivo keeping bone formation and resorption coupled. This same paper showed results of 3AC SHIP1 inhibitor in vivo.

How do you interpret your studies in this context where your mice have deletion in both osteoclasts and mesenchymal stem cells plus minus lymphoid cells? Have you looked at osteoblast cultures from Rag2^{-/-}/Il2rg^{-/-}/SHIP1^{styx/styx} and the other strains? Histomorphometry with dynamic bone formation?

Additional discussion should include how to interpret the transcriptomics with the knowledge that SHIP1 plays a major role in osteoblasts in vivo. Are the changes you find do to reduction of inflammation influencing osteoblasts?

The experiments appear to be well controlled and convincing except as mentioned below.

1. Figure 3g: the X axis is mislabeled and does not include 5 and 10um
2. Figure 3H and I: Day 5 mature osteoclasts, especially very large ones like those seen in SHIP1 KO cultures, are very adherent, fragile and often do not survive lifting and spinning down. Recommend to confirm that osteoclasts are present on the calcium phosphate substance after 24 hours by staining them or use day 3 osteoclasts and let the go 24-48 hours. Additionally, use of equal microliters of cells instead of equal numbers of cells leave error especially if the large osteoclasts did not survive replating.
3. Figure S3. It is very consistent that Rag2^{-/-}/Il2rg^{-/-}/SHIP1^{+/+} had significantly decreased SHIP1 expression in the bone marrow. Why is it reduced? Any speculation?
4. Figure S6 C and D. The Y axis is so different it is really difficult to compare results in C to D. Please make the Y axis similar.
5. Perhaps mention that increased TRAP at zero RANKL likely indicates activated macrophages.

Reviewer #2 (Comments to the Authors (Required)):

This manuscript by Safari et al. explores the role of SHIP1 in skeletal homeostasis. This study aims to determine the role of the

immune environment in the background of the loss of SHIP expression. SHIP is a negative regulator of the PI3K pathway, a pathway critical for osteoclast differentiation. Previous studies had demonstrated that SHIP knockout mice have reduced bone mass and a change in skeletal architecture. Lymphocyte deletion rescued the skeletal phenotype. Interestingly the phenotype of osteoclasts was not change with lymphocyte deletion. Even though this is a well thought out study and adds to the increasing knowledge about the role of inflammation and skeletal maintenance, there are some major and minor points that should be addressed by the authors to make the findings more significant.

1. The authors need to correlate proliferation with differentiation. It has been demonstrated that osteoclasts must exit proliferation to fuse to become multinuclear. As it stands now there is no clear connection between the changes in proliferation and the enhancement of differentiation. Authors should indicate in Sup Fig 4 what concentration was used for differentiation. If possible differentiation should be repeated with different amounts of RANKL (as was used in proliferation assay) to correlate proliferation and differentiation.
2. If possible decalcified bone sections from mice in the Rag2^{-/-}; Il2rg^{-/-}; SHIP^{styx/styx} background should be TRAP stained to demonstrate that the osteoclast phenotype seen in vitro is also seen in vivo.

Minor Points:

1. In the results section an introduction/explanation and reference should be given on the SHIP^{styx/styx} mouse model.
2. In Figure 1 the data from the Rag2; Il2rg different backgrounds should be separated to make it easier for the reader to understand.
3. Number and sex of mice analyzed should be included in figure legends.

**Point-by-point reply to the Reviewers' comments**

**General comments**

Manuscript sections where text has been added, modified or removed compared with the
previously submitted version have been underlined in the revised version of the manuscript.
We provide both a version of the revised manuscript with these changes indicated and a version
without indication of changes.

**Comments of Reviewer #1:**

***Reviewer #1 / Major comments:***

*1. The conclusions are mostly supported by the data. However, discussion is lacking in terms*
*of putting these results in the context of mesenchymal stem cell-specific and myeloid-specific*
*SHIP1 KO mice published by Iyer et. all in 2014. Prior studies by Iyer et. all, showed that mice*
*with myeloid cell specific deletion of SHIP1 in osteoclast precursors had normal bone mass*
*and osteoclasts in vivo but substantially larger TRAP+ osteoclasts in vitro. They also showed*
*that mice with mesenchymal stem cell-specific (OSXCre) depletion of SHIP1 have runt size,*
*low bone mineral apposition and low bone mass in the absence of inflammation. Importantly,*
*they showed that MSC-SHIP1 deficient mice also had lower osteoclast numbers in vivo and in*
*vitro. They conclude that SHIP1 expression in the mesenchymal stem cell compartment*
*provides a negative regulation of osteoclastogenesis in vivo keeping bone formation and*
*resorption coupled. This same paper showed results of 3AC SHIP1 inhibitor in vivo.*
*How do you interpret your studies in this context where your mice have deletion in both*
*osteoclasts and mesenchymal stem cells plus minus lymphoid cells?*

We thank Reviewer #1 for these comments. First, the study Reviewer #1 refers to (Iyer *et al.*;
PMID: 25525673) indicates only a phenotype in mice with SHIP1-deficiency in mesenchymal
stem and progenitor cells (MS/PC; *Osx-Cre/SHIP^{lox/lox}* mice), yet not in animals lacking
SHIP1 in more mature osteoblasts (*Col1a1-Cre/SHIP^{lox/lox}* mice) – where it is probable that
only a subset of the osteoblast lineage cells lacks SHIP1. Iyer *et al.* reported the *Osx-*
*Cre/SHIP^{lox/lox}* mice to harbour specific ablation of SHIP1 in MS/PC. However, a recent study
has demonstrated expression of *Osx/Sp7* in a subset of hematopoietic stem cells and several

hematopoietic progenitor subtypes, including several multipotent progenitor (MPP) subsets
(MPP1-4) as well as in common lymphoid progenitors (CLP) (PMID: 32755539). Therefore,
it is expected that, in addition to MS/PC, some hematopoietic cells also lack SHIP1 expression
in *Osx-Cre/SHIP^{lox/lox}* mice. Further, transgenic Cre expression *per se* in *Osx-Cre* mice leads
to decreased body weight after postnatal day 7 – which persists until up to 12 weeks of age –
and these mice show transient bone phenotypes (PMID: 22160436; 25139670; 25550101).
Notably, Iyer *et al.* did not use these *Osx-Cre* mice as controls for *Osx-Cre/SHIP^{lox/lox}* animals,
but rather *SHIP^{lox/lox}* mice. In addition, Iyer *et al.* mentioned *Osx-Cre/SHIP^{lox/lox}* mice show
growth defects and remain smaller than controls from weaning on. Importantly, we
demonstrate in Figure S1 of the manuscript that, for both femora and vertebrae, BV/TV and
trabecular thickness are correlated with the age and size of mice. Therefore, a direct comparison
of two groups of mice of different sizes will ineluctably lead to changes in these parameters.
In summary, the multifaceted nature of *Osx-Cre/SHIP^{lox/lox}* mice complicates the interpretation
of their phenotype.

Second, the data presented in our manuscript suggest a minor role, if at all, of SHIP1 in the
development and maintenance of the skeleton. Indeed, *SHIP1^{styx/styx}* mice are entirely KO for
SHIP1; therefore, both their hematopoietic and mesenchymal compartments are deficient in
SHIP1. The fact that *Rag2^{-/-}/Il2rg^{-/-}/SHIP1^{styx/styx}* mice develop normally and show normal
skeletal development, suggests that the phenotype in *SHIP1^{styx/styx}* mice is caused by the
systemic inflammation triggered by lack of SHIP1 in hematopoietic cells of these animals. We
thus conclude that SHIP1 deficiency in either the hematopoietic osteoclastogenic or the
mesenchymal osteogenic cell lineages is not sufficient to induce a bone phenotype in non-
inflammatory or unchallenged conditions, which we also discuss in the manuscript (lines 307-
310 in submitted .docx file).

Third, we have not performed any experiments including cultures of osteoblast-like cells.
SHIP1-deficient *SHIP1^{styx/styx}* mice have a limited lifespan and have to be generated by
intercrossing *SHIP1^{+ /styx}* parents. The difficult breeding of this strain has prevented us from
getting enough genotyped pups (1-2 days old) to prepare osteoblasts (OB) from *SHIP1^{styx/styx}*,
*SHIP1^{+ /styx}* and WT mice and evaluate the contribution of these cells in our model.

It will be the subject of future studies to address the question as to whether SHIP1 deficiency
may lead to a bone phenotype in challenged, pathophysiological settings, and to assess whether
under such conditions a possible phenotype is produced by the osteoclast or by the OB cell
lineage.

2. Have you looked at osteoblast cultures from *Rag2*^{-/-}/*Il2rg*^{-/-}/*SHIP1*^{styx/styx} and the other
strains? Histomorphometry with dynamic bone formation?

We have not looked at *Rag2*^{-/-}/*Il2rg*^{-/-}/*SHIP1*^{styx/styx} OB as suggested by Reviewer #1. While
perfectly viable, *Rag2*^{-/-}/*Il2rg*^{-/-}/*SHIP1*^{styx/styx} mice produce small sized litters (the reasons for it
being unknown) and need to be bred from intercrossed *Rag2*^{-/-}/*Il2rg*^{-/-}/*SHIP1*^{+ /styx} parents.
Therefore, for this strain also, these logistic constraints in the breeding do not allow us to
produce the number of pups necessary to isolate calvarial OB lineage cells.

We did, however, perform experiments with primary OB from WT C57BL/6J mice, which
were cultured in media supplemented with the SHIP1 inhibitor 3AC. Our data indicate that
pharmacologic blockade of SHIP1 caused a reduction in cell viability at concentrations ≥ 8.5
nM. At the same time, the ratio of ALP/XTT did not change in these cells, demonstrating a
primary effect of SHIP1 on cell proliferation rather than differentiation, as determined by ALP
activity – ALP being a parameter of OB differentiation (Fig. R1A).

On the other hand, OB activity appears to be attenuated in mice lacking SHIP1 compared to
SHIP1-proficient controls. Indeed, we found in total RNA from diaphyseal bone of SHIP1-
deficient strains (with *Rag2*^{-/-}/*Il2rg*^{-/-} and *Rag2*^{+ /+}/*Il2rg*^{+ /+} backgrounds) a reduction in
transcript levels of *Coll*, the major component of bone extracellular matrix that is produced by
OB (Fig. R1B).

We do agree with Reviewer #1 that the data presented herein, which are not part of the
manuscript, do not fully explain the role of OB lineage cells in *SHIP1*^{styx/styx} strains. Part of the
answer to the question by Reviewer #1 may be obtained by performing *in vivo* fluorescence
labeling of the bones to assess bone apposition rate. These investigations will be performed in
the future for the strains used in the study.

*3. Additional discussion should include how to interpret the transcriptomics with the*

*knowledge that SHIP1 plays a major role in osteoblasts in vivo. Are the changes you find*

*to reduction of inflammation influencing osteoblasts?*

Even though we did not find evidence for a major role of SHIP1 in osteoblasts in vivo, we did

check whether the expression of OB-associated genes is dependent on SHIP1, a list of genes

characterizing the OB phenotype was composed, and a heatmap for these OB genes was

generated using the data from the transcriptome analysis of the femora, as we did in the

manuscript for OC cells.

Fig. R2. Heatmap of selected osteoblast-associated gene markers. This transcriptome analysis was generated using the RNA sequencing data from the femoral diaphysis of the indicated mouse strains. *Fgf23* (red arrow) is specifically upregulated in the bone tissue of *Rag2*^{+/+}/*Il2rg*^{+/+}/*SHIP1*^{styx/styx} mice.

Our data indicate that several of the selected genes associated with an OB phenotype were
 downregulated in *SHIP1*^{styx/styx} mice, independently of the expression of *Rag2*^{-/-} and *Il2rg*^{-/-} (Fig.
 R2). Thus, osteoblastic gene expression appears, in general, to be modulated by SHIP1.
 However, loss of lymphoid cells partially compensated the phenotype observed in SHIP1-
 deficient mice, as observed for levels of transcripts encoding *Fgf23* (a factor controlling
 phosphatemia), which were specifically upregulated in *Rag2*/*Il2rg*-proficient *SHIP1*^{styx/styx}
 animals. This is in line with the findings we described in the manuscript for the transcriptomic
 signature of *SHIP1*^{styx/styx} versus *Rag2*^{-/-}/*Il2rg*^{-/-}/*SHIP1*^{styx/styx} OC, compared to their respective
 controls (Figure 3 of the manuscript).

*4. The experiments appear to be well controlled and convincing except as mentioned below:*
 *Figure 3g: the X axis is mislabeled and does not include 5 and 10um*

We thank Reviewer #1 for pointing this out. We have corrected the information displayed in
 this Figure (Figure 2G).

5. Figure 3H and I: Day 5 mature osteoclasts, especially very large ones like those seen in
 SHIP1 KO cultures, are very adherent, fragile and often do not survive lifting and spinning
 down. Recommend to confirm that osteoclasts are present on the calcium phosphate substance
 after 24 hours by staining them or use day 3 osteoclasts and let them go 24-48 hours.
 Additionally, use of equal microliters of cells instead of equal numbers of cells leave error
 especially if the large osteoclasts did not survive replating.

This is a pertinent comment from Reviewer #1 pointing out potential complications for the
 resorption assay used in this study. First, for the assays quantifying OC activity in the
 manuscript (i.e. resorption that is measured by quantifying ^{45}Ca release from the calcium
 phosphate layer), the culture period was restricted to 24h to limit the effects of OC development
 and to focus on OC activity. But we are aware that OC differentiation takes place also during
 this 24h culture period. Second, in this resorption experiment, we also determined TRAP
 activity in OC as a parameter for the presence of these cells after adhesion and after the 24h
 experiment. Third, in individual control experiments, we performed TRAP staining to visualize
 OC (Fig. R3). Lastly, ^{45}Ca release was normalized to the TRAP activity to correct for
 differences in cell numbers. Altogether, we think that the design of our experiments is such to
 take into account the concerns of Reviewer #1.

**Fig. R3. TRAP-stained OC from the indicated strains after 24h of subculture on plastic culture dish.** OPC
 cultures were treated with 20 ng/ml RANKL for 5 days. Thereafter, they were sub-cultured on 24-well plates.
 TRAP staining was performed after 24h to visualize OC.

6. Figure S3. It is very consistent that *Rag2*^{-/-}/*Il2rg*^{-/-}/*SHIP1*^{+/+} had significantly decreased
*SHIP1* expression in the bone marrow. Why is it reduced? Any speculation?

We agree with this comment from Reviewer #1 and we can not provide a direct explanation
for this finding besides the following comments:

First, these results show transcript levels as determined by PCR analysis from *in vitro* generated
OC; therefore, these data should not be affected by the obvious changes present in the bone
marrow composition of *SHIP1*-proficient *Rag2*^{-/-}/*Il2rg*^{-/-} versus WT mice.

Second, the lack of expression of *Il2rg* (encoding for the common cytokine receptor gamma
chain) in *Rag2*^{-/-}/*Il2rg*^{-/-} mice possibly interferes with the expression of *SHIP1*. However, we
did not pay too much attention to this finding since the data in this same figure also indicate
that 50% levels of transcripts encoding for *SHIP1* are sufficient to correct in heterozygous
*Rag2*^{+/+}/*Il2rg*^{+/+}/*SHIP1*^{+/*styx*} mice the phenotype of homozygous *SHIP1*^{*styx/styx*} animals (the *styx*
mutation is a recessive trait).

7. Figure S6 C and D. The Y axis is so different it is really difficult to compare results in C to
D. Please make the Y axis similar.

We improved the figure correspondingly.

8. Perhaps mention that increased TRAP at zero RANKL likely indicates activated
macrophages.

We consider TRAP signal at zero RANKL an experimental background. Even though we assay
for TRAP, there may be other acid phosphatases that are not fully blocked by tartrate or give
some background under the conditions of our assay.

Please also refer Supplementary Figure 4 in the manuscript, which indicates negligible
(normalized) TRAP activity in OPC cultured without supplementation of RANKL, when
compared to conditions where RANKL was added.

**Comments of Reviewer #2:**166 **Reviewer #2 / Major comments:**

*1. The authors need to correlate proliferation with differentiation. It has been demonstrated*
*that osteoclasts must exit proliferation to fuse to become multinuclear. As it stands now there*
*is no clear connection between the changes in proliferation and the enhancement of*
*differentiation. Authors should indicate in Sup Fig 4 what concentration was used for*
*differentiation. If possible differentiation should be repeated with different amounts of RANKL*
*(as was used in proliferation assay) to correlate proliferation and differentiation.*

Reviewer #2 raises a relevant point that we have extensively discussed among the authors. We
decided, for each RANKL concentration, to set to 100% the TRAP values of the WT group to
better visualize the TRAP values of the other genotypes. Yet it is correct that such
representation of the data does not inform on the number of OC that are generated in
dependence of TRAP concentration. Therefore, we have included this data in the manuscript,
as a supplementary Figure (Figure S4).

*2. If possible decalcified bone sections from mice in the Rag2^{-/-}; Il12rg^{-/-}; SHIP^{styx/styx}*
*background should be TRAP stained to demonstrate that the osteoclast phenotype seen in vitro*
*is also seen in vivo.*

We performed TRAP staining on bone tissues of L4 vertebrae of different mouse genotypes
used in this study. No obvious changes in the size and number of OC were observed in bones
from SHIP1-deficient SHIP1^{styx/styx} versus control mice (Fig. R4). Therefore, although
SHIP1^{styx/styx} OC proliferate more *in vitro* cultures, the number of OC is similar in bone tissues
from these animals as compared to WT animals. This observation is in line with the
interpretation of our data that the phenotype in SHIP1^{styx/styx} mice is caused by the systemic
inflammation triggered by lack of SHIP1 in the hematopoietic lineage of these animals, rather
than intrinsic changes in the number and/or function of their OC or OB.

**Reviewer #2 / Minor comments:**

*5. 1. In the results section an introduction/explanation and reference should be given on the*
*SHIP^{styx/styx} mouse model.*

This information was added to the revised version of the manuscript (lines 128-130 in
submitted .docx file).

*2. In Figure 1 the data from the Rag2; Il2rg different backgrounds should be separated to*
*make it easier for the reader to understand.*

We have modified this Figure accordingly.

*3. Number and sex of mice analyzed should be included in figure legends.*

We added the information on the sex of the animal used in the Material and Methods (lines
398-399 in submitted .docx file; “For *ex vivo* and *in vitro* experiments, mice were age- and sex-
matched”). In addition, information on the number of mice used to generate the data is now
indicated in the figure legends.

January 9, 2024

RE: Life Science Alliance Manuscript #LSA-2023-02297-TR

Prof. Philippe Krebs
Institute of Tissue Medicine and Pathology, University of Bern
Murtenstrasse 31
Bern CH - 3008
Switzerland

Dear Dr. Krebs,

Thank you for submitting your revised manuscript entitled "SHIP1 deficiency causes inflammation-dependent retardation in skeletal growth". We would be happy to publish your paper in Life Science Alliance pending final revisions necessary to meet our formatting guidelines.

- please address Reviewer 1's remaining points #2 and 3
- please be sure that the authorship listing and order is correct
- we encourage you to revise the figure legends for figures S3 and S7 such that the figure panels are introduced in an alphabetical order

A. FINAL FILES:

B. MANUSCRIPT ORGANIZATION AND FORMATTING:

Thank you for your attention to these final processing requirements. Please revise and format the manuscript and upload materials within 21 days.

Sincerely,

Reviewer #1 (Comments to the Authors (Required)):

While the authors have addressed reviewer comments in a rebuttal, they have not significantly improved the paper and major questions still remain.

1) Because there are changes in OB and OC, histomorphometry is required to fully understand the in vivo skeletal phenotype. The histology provided in rebuttal indicates that the OC are larger and likely have increased trap+ bone surface in the SHIP styx/sytx in vivo. Also we need the RAG-*IL2gR*^{-/-}-Styx/styx vertebrae trap staining histology. True quantification of osteoclast numbers and trap+ bone surface, Eroded surface and OB numbers are needed.

2) TRAP is not a good measure of number of osteoclasts in the SHIP^{-/-} cultures. Real quantification by counting the number of the multinucleated, trap+ osteoclasts over the RANKL concentrations is required to definitely address if SHIP^{-/-} OC are less resorptive and make fewer OC. The area of osteoclast should also be quantified across the genotypes.

3) In rebuttal figures, the authors did not include RAG-*IL2gr*^{-/-} background mice for comparison. This is critical and this data should be included in a manuscript.

Reviewer #2 (Comments to the Authors (Required)):

The authors have addressed all my concerns and I do not have any additional comments.

**Point-by-point reply #2 to the Reviewers' comments**

**General comments**

Manuscript sections where text has been added, modified or removed compared with the
previously submitted version have been underlined in the revised version of the manuscript.
We provide both a version of the revised manuscript with these changes indicated and a version
without indication of changes.

**Editorial comments:**

*We would be happy to publish your paper in Life Science Alliance pending final revisions*
*necessary to meet our formatting guidelines. Along with points mentioned below, please tend*
*to the following:*

*1. please address Reviewer 1's remaining points #2 and 3*

We have addressed below comment #2 and comment #3 of Reviewer #1.

*2. please be sure that the authorship listing and order is correct*

We have cross-checked the authorship listing and the different affiliations.

*3. we encourage you to revise the figure legends for figures S3 and S7 such that the figure*
*panels are introduced in an alphabetical order*

We have correspondingly revised the legends of Figures S3 and S7.

**Reviewer #1 / Major comments:**

*2. TRAP is not a good measure of number of osteoclasts in the SHIP^{-/-} cultures. Real*
 *quantification by counting the number of the multinucleated, trap⁺ osteoclasts over the*
 *RANKL concentrations is required to definitely address if SHIP^{-/-} OC are less resorptive and*
 *make fewer OC. The area of osteoclast should also be quantified across the genotypes.*

As requested by Reviewer #1, we went back to the original experiments and counted osteoclast
 (OC)-like cells to compare the findings between the number of TRAP⁺ cells (with ≥ 3 nuclei)
 and TRAP activity and to standardize resorption activity against OC number.

As displayed in Fig. R1, these new data are comparable to the findings shown in Fig. 2 of the
 manuscript. In particular, the number of TRAP⁺ cells was found to be proportional to the
 applied concentration of RANKL, yet without consistent difference between mouse genotypes
 (Fig. R1 A, B) – which aligns with the corresponding findings of manuscript Fig. B and E
 indicating TRAP activity. In addition, we found a mild but consistent trend toward a reduced
 ratio of TRAP⁺ cells/XTT while directly comparing WT versus *SHIP1^{styx/styx}* groups (Fig. R1
 C), which was not observable in *Rag2^{-/-}/Il2rg^{-/-}* strains (Fig. R1 D). Importantly, TRAP⁺ OC
 from SHIP1-deficient strains consistently showed a reduced ability to resorb calcium while
 cultured in the presence of higher concentrations of RANKL (≥ 3 ng/ml) (trend in Fig. R1 E
 and significant difference in Fig. R1 F), which further corroborates the findings shown in the
 manuscript Fig. 2H, I. We nevertheless believe that quantifying total TRAP in cell lysates
 presents several advantages over the (semi-quantitative) enumeration of OC:

- • In general, counting of OC exhibits a larger variation than does TRAP determination
 (see Fig. R1 and also the line of argumentation below).
- • For counting, OC are defined as TRAP-positive cells with 3 or more nuclei; smaller
 cells of the OC lineage are not included. This is also due to a practical reason since
 smaller cells would be too numerous to be counted by eye (see Fig. R2).
- • Counting OC has an intrinsic bias and may be dependent on the experimenter counting
 the cells. Furthermore, certain experimenters may only count the cells in a portion of a
 culture well, or in a few wells of the culture dish, and then extrapolate the counts for an
 entire well or the other wells. Note that for the data shown in Fig. R1, whole wells were
 counted (for a total of up to ca. 1000 TRAP⁺ cells per well for certain genotypes and in
 cultures with 20ng/ml RANKL), with two wells per condition.

• Therefore, assessing TRAP activity in the cell lysate of an entire cell culture well or
plate represents a quantitative – and not semi-quantitative – method that effectively
mitigates potential biases introduced by the experimenter or by the cell counting in
partial areas.

• Lastly, small OC may resorb only small pits that are not visible if the resorption assay
is carried out on a bone-like substrate (i.e. dentin wavers, bovine bone, ivory wavers,
or others). In the resorption assay used in our study, we measured solubilized ⁴⁵Ca from
the calcium phosphate (CaP) layer, which corresponds to the total solubilization of the
mineral in a particular well. For such kind of assay, the contribution of the small OC
lineage cells must be considered, which we also included in our measure by determining
total TRAP activity.

Considering the results presented in Fig. R1, provided in response to Reviewer #1's request,
we maintain that it is justified to include in the manuscript the more comprehensive dataset
incorporating TRAP activity.

**Fig. R1. Quantification of TRAP⁺ OC and OC resorption activity in SHIP1-deficient mice.** Osteoprogenitor
 cells (OPC) from WT; *SHIP1*^{+/styx}; *SHIP1*^{styx/styx} (A, C, E) and *Rag2*^{-/-}/*Il2rg*^{-/-}/*SHIP1*^{+/+}; *Rag2*^{-/-}/*Il2rg*^{-/-}/*SHIP1*^{+/styx};
 and *Rag2*^{-/-}/*Il2rg*^{-/-}/*SHIP1*^{styx/styx} (B, D, F) mice were cultured in media supplemented with CSF-1 and various
 concentrations of RANKL. After 5 days in culture, the number of TRAP⁺ cells (with ≥ 3 nuclei) (A, B) and the
 number of TRAP⁺ cells/XTT (C, D) were counted. Alternatively, the mineral dissolution activity of TRAP⁺ cells
 was measured after transfer of OC onto a layer of amorphous CaP spiked with ⁴⁵Ca. Data indicate mean ± SD of
 three independent experiments, each with one mouse per group as a cell donor. For E and F, the number of TRAP⁺
 cells was extrapolated from a distinct experiment applying the same experimental conditions. Significant changes
 were calculated with two-way ANOVA, **(*p* < 0.01), ***(*p* < 0.001), and ****(*p* < 0.0001).

3. In rebuttal figures, the authors did not include *RAG*^{-/-}*IL2gr*^{-/-} background mice for
 comparison. This is critical and this data should be included in a manuscript.

We provide below TRAP staining photographs of all genotypes analyzed in this study. Note
 that OC from all *Rag2*^{-/-}*Il2rg*^{-/-} strains formed larger monocellular aggregates compared OC
 from *Rag2*^{+/+}*Il2rg*^{+/+} strains (Fig. R2). We have no explanation for this phenomenon, which
 may relate to the lack of expression of *Il2rg* (encoding for the common cytokine receptor
 gamma chain). Therefore, we prefer not to include these data in the manuscript, as it would
 unnecessarily complicate the main message of our findings without contributing substantial
 novel information.

**Fig. R2. TRAP-stained OC from the indicated strains after 1h adhesion on tissue culture plates.** OPC
 cultures were treated with 20 ng/ml RANKL for 5 days. Thereafter, cells were released and transferred into 96-
 well tissue culture plates. TRAP staining was performed after an adhesion period of 1h to visualize adherent OC.

However, we provide in the new revised version of the manuscript pictures of TRAP staining
 on bone tissues of L4 vertebrae for all studied mouse strains. These data corroborate our
 previous findings by indicating no obvious changes in the size and number of OC in bone
 tissues from SHIP1-deficient versus control strains (Fig. S6 in the revised manuscript).

February 6, 2024

RE: Life Science Alliance Manuscript #LSA-2023-02297-TRR

Prof. Philippe Krebs
Institute of Tissue Medicine and Pathology, University of Bern
Murtenstrasse 31
Bern CH - 3008
Switzerland

Dear Dr. Krebs,

Thank you for submitting your Research Article entitled "SHIP1 deficiency causes inflammation-dependent retardation in skeletal growth". It is a pleasure to let you know that your manuscript is now accepted for publication in Life Science Alliance. Congratulations on this interesting work.

DISTRIBUTION OF MATERIALS:

Again, congratulations on a very nice paper. I hope you found the review process to be constructive and are pleased with how the manuscript was handled editorially. We look forward to future exciting submissions from your lab.

Sincerely,
